# Analysis of economic and educational spillover effects in PEPFAR countries

William Crown[1]*, Dhwani Hariharan[1], Jennifer Kates[2], Gary Gaumer[1], Monica Jordan[1], Clare Hurley[1], Yiqun Luan[1], Allyala Nandakumar[1]

**1** Institute for Global Health and Development, Heller School for Social Policy and Management, Brandeis University, Waltham, MA, United States of America, **2** KFF, Washington, DC, United States of America

\* wcrown@brandeis.edu

**Data Availability Statement:** Our data came from four publicly available datasets: World Bank's World Development Indicators; U.S. government's foreignassistance.gov database; OECD Creditor Reporting System database; and the Institute of

## Abstract

The United States President's Emergency Plan for AIDS Relief (PEPFAR) has been credited with saving millions lives and helping to change the trajectory of the global human immunodeficiency virus (HIV) epidemic. This study assesses whether PEPFAR has had impacts beyond health by examining changes in five economic and educational outcomes in PEPFAR countries: the gross domestic product (GDP) per capita growth rate; the share of girls and share of boys, respectively, who are out of school; and female and male employment rates. We constructed a panel data set for 157 low- and middle-income countries between 1990 and 2018 to estimate the macroeconomic impacts of PEPFAR. Our PEPFAR group included 90 countries that had received PEPFAR support over the period. Our comparison group included 67 low- and middle-income countries that had not received any PEPFAR support or had received minimal PEPFAR support (<$1M or <$.05 per capita) between 2004 and 2018. We used differences in differences (DID) methods to estimate the program impacts on the five economic and educational outcome measures. This study finds that PEPFAR is associated with increases in the GDP per capita growth rate and educational outcomes. In some models, we find that PEPFAR is associated with reductions in male and female employment. However, these effects appear to be due to trends in the comparison group countries rather than programmatic impacts of PEPFAR. We show that these impacts are most pronounced in COP countries receiving the highest levels of PEPFAR investment.

## Introduction

A large literature has demonstrated that health investments are correlated with educational attainment and economic growth [1–7]. However, analysis of the economic and educational impacts of health investments made by specific programs is less common. The United States President's Emergency Plan for AIDS Relief (PEPFAR) is the largest commitment by any country addressing a single disease [8–11]. PEPFAR has been credited with saving 25 million lives and helping to change the trajectory of the global human immunodeficiency virus (HIV) epidemic [12]. In prior analyses, we found that PEPFAR has contributed to large, significant reductions in all-cause mortality, suggesting a mortality effect beyond HIV [13], as well as

Health Metrics and Evaluation GBD Result's Tool. The URLs to these dataset are as follows: 1. World Bank's World Development Indicators: https://datatopics.worldbank.org/world-development-indicators/ 2. US Department of State, USAID: https://www.foreignassistance.gov/ 3. OECD Creditor Reporting System database: https://stats.oecd.org/Index.aspx?DataSetCode=crs1 4. Institute of Health Metrics and Evaluation GBD Result's Tool: https://ghdx.healthdata.org/gbd-2019, and http://ghdx.healthdata.org/gbd-results-tool.

**Funding:** This paper was produced with funding from Palladium International, LLC under subcontract number 217730-Brandeis-01; and Prime Contract number 2021-002516 from the Global Fund to Fight AIDS, Tuberculosis, and Malaria. Its contents are solely the responsibility of Brandeis University and do not necessarily represent the official views of Palladium or The Global Fund.The funders had no role in study design, data collection and analysis, decision to publish, or preparation of the manuscript.

**Competing interests:** The authors have declared that no competing interests exist.

significant, positive, health spillover effects in the area of maternal and child health, including reductions in maternal and child mortality and increases in childhood immunization rates [13].

In this analysis, we seek to assess whether PEPFAR has had impacts beyond health by examining changes in five economic and educational outcomes in PEPFAR countries: the gross domestic product (GDP) per capita growth rate; the share of girls and share of boys, respectively, who are out of school; and female and male employment rates. Since its launch in 2003, PEPFAR has provided approximately $90 billion in bilateral assistance to address HIV in low- and middle-income countries (LMICs) to provide services directly and to purchase supplies, local labor, real-estate, utilities, and various contracted services. While PEPFAR, as an HIV-focused and targeted effort, was not designed to be an economic or educational program, there are several reasons to think that such spending could potentially have positive externalities for the economy and on educational attainment. Barofsky and Nosair (2015) enumerate three basic economic benefits of improved population health: "1) greater labor productivity and school attendance from less absenteeism, 2) better cognition and school performance through less disease in utero and in early life, and 3) greater incentives for education and savings with lengthened life expectancy" [14]. Even more directly, program impacts on mortality and morbidity in the population would be expected to have positive effects on labor supply.

In addition, since its inception, PEPFAR has operated programs designed to help orphans and vulnerable children (OVC), which include a focus on socio-economic factors and facilitate access not just to health programs but also social, legal, and economic support [15]. Further, over time, PEPFAR has incorporated interventions that include economic and educational support, such as in its DREAMS program focused on adolescent girls and young women that addresses the drivers of the HIV epidemic [16–18]. In addition, external aid may also act as a direct economic stimulus in countries, impacting their GDP [19].

This analysis aims to add to the limited research and evidence on such effects. A study published in 2015 showed that PEPFAR investments led to increases in male employment in ten PEPFAR-focus countries but did not show similar results for female employment [19]. A paper published in 2017 showed that PEPFAR investments contributed positively to GDP growth rates [19]. Similarly, the Bipartisan Policy Center found that GDP per capita and productivity per worker were positively correlated with the level of PEPFAR investments [20]. There are however no studies that have looked at PEPFAR investments and educational attainment.

For the current analysis, we look at a larger set of countries and over a longer period of time than the prior analyses identified. We use a difference-in-difference quasi-experimental design to analyze the change in each of these outcomes in 90 PEPFAR countries between 2004, the first year in which PEPFAR funding began, and 2018, compared to a comparison group of 67 low- and middle-income countries (See methodology for more detail). We tested several different model specifications. Our final model controls for numerous baseline variables that may also be expected to influence these outcomes and which help to make the PEPFAR and non-PEPFAR country groups more comparable.

## Methods

We constructed a panel data set for 157 low- and middle-income countries between 1990 and 2018 to estimate the macroeconomic impacts of PEPFAR. Our PEPFAR group included 90 countries that had received PEPFAR support over the period. Our comparison group included 67 low- and middle-income countries that had not received any PEPFAR support or had received minimal PEPFAR support (<$1M or <$.05 per capita) between 2004 and 2018. Data

**Table 1. Baseline variables.**

| Variable | Data Source |
|---|---|
| 1. GDP per capita (current USD) | World Bank Development Indicators, [22] |
| 2. Recipient of U.S. HIV funding prior to 2004 (dummy variable) | USAID, [21] |
| 3. Total population | United Nations, Department of Economic and Social Affairs, Population Division, [23] |
| 4. Life expectancy at birth (years) | World Bank Development Indicators, [22] |
| 5. Total fertility rate (births per woman) | World Bank Development Indicators, [22] |
| 6. Percent urban population (of total population) | World Bank Development Indicators, [22] |
| 7. School enrollment, secondary (% gross) | World Bank Development Indicators, [22] |
| 8. WB country income classification | World Bank, [24] |
| 9. HIV prevalence (% of population ages 15–49) | World Bank Development Indicators, [22]<br>To address missing values in some cases, additional data were obtained from the Global Burden of Disease Collaborative Network, [25] |
| 10. Per capita donor spending on health (non-PEPFAR) (constant $) | OECD Creditor Reporting System database, [26] |
| 11. Per capita domestic health spending, government and private, PPP (current $) | World Bank Development Indicators, [22] |

Notes: GDP = gross domestic product; HIV = human immunodeficiency virus; OECD = Organization for Economic Cooperation and Development; PEPFAR = US President's Emergency Plan for AIDS Relief; PPP = purchasing power parity; USAID = United States Agency for International Development; WB = World Bank; WDI = world development indicators.

on PEPFAR spending by country were obtained from the U.S. government's https://foreignassistance.gov/ database [21] and represent U.S. fiscal year disbursements. The baseline variables are reported in Table 1.

Impact estimates of PEPFAR are obtained with a difference-in-differences econometric model that utilizes PEPFAR participation beginning in 2004. Impact estimates are made for all PEPFAR recipient countries as a group, as well as for the 31 countries that submitted Country Operational Plans (COPs) during the period. The largest Impacts of PEPFAR would be expected in the COP countries because they received the largest funding amounts and country teams were actively engaged in the planning process for the investment of program dollars. The comparison group of LMICs includes 46 unfunded countries and 18 minimally funded countries.

We estimate impacts of PEPFAR on five economic and educational outcome measures including the GDP per capita growth rate, percentage of female adults employed, percentage of male adults employed, girl's educational disengagement (ratio of primary school age females out of school to the population of primary school age females), and boy's educational disengagement (ratio of primary school age males out of school to the population of primary school age males).

DID methods have been widely used in the program evaluation literature to estimate treatment effects as a non-parametric alternative to parametric sample selection models [27]. DID can be thought of as an extension of quasi-experimental design to account for unobserved variables potentially correlated with both an intervention and the outcome that are assumed to remain fixed over time. The method can be used when two periods of data are available for

countries that receive an intervention (in this case, PEPFAR funding) and those that do not (the comparison group). In the baseline period, PEPFAR countries have not yet received any PEPFAR program dollars (although they may have received external HIV funding, which we control for, as described below). Characteristics of the comparison group countries are also measured in the baseline period. The first group of PEPFAR countries began receiving funding in 2004 and their outcomes are observed in the second (follow-up) period. We also measure the outcomes for countries in the comparison group in the same follow-up period. If we assume that countries may also have unobserved characteristics, λi, that are correlated with outcomes but that these characteristics remain fixed over time (e.g., unobserved health endowment), DID provides a method to control for these fixed, unobserved characteristics. The outcome equations for periods 1 and 2 are shown in Eqs 1A and 1B, respectively:

$$Y_{i1} = B_0 + B_1 X_{i1} + B_2 \lambda_i + \epsilon_{i1} \tag{1A}$$

$$Y_{i2} = B_0 + B_1 X_{i2} + B_2 \lambda_i + B_3 T_i + \epsilon_{i2} \tag{1B}$$

Calculating the change in outcomes and explanatory variables between time 1 and time 2, and re-estimating the outcome equation, is equivalent to subtracting Eqs (1A) from (1B):

$$(Y_{i2} - Y_{i1}) = (B_0 - B_0) + B_1(X_{i2} - X_{i1}) + B_2(\lambda_i - \lambda_i) + B_3 T_i + (\epsilon_{i2} - \epsilon_{i1}) \tag{2}$$

Which simplifies to:

$$(Y_{i2} - Y_{i1}) = B_1(X_{i2} - X_{i1}) + B_3 T_i + (\epsilon_{i2} - \epsilon_{i1}) \tag{3}$$

In other words, the DID approach subtracts out unobserved fixed effects of countries that may be correlated with both treatment selection and outcomes.

Operationally, the DID model is easy to implement and generates three key parameter values of interest. A time dummy variable captures the overall differences in the mean value of the dependent variable between the baseline period and the follow-up period for the comparison group. A dummy variable = 1 for PEPFAR countries and 0 for comparison group countries and measures the differences between the two groups prior to the intervention. Finally, the coefficient on the variable representing the interaction between PEPFAR and the time dummy variables measures the program impact of PEPFAR.

The countries that received substantial PEPFAR support during 2004 to 2018 were not a random sample of LMICs. As a result, we also estimate DID models controlling for several covariates to achieve better balance with the comparison group. These covariates include the urban population percentage, HIV prevalence rate, life expectancy, whether the US had provided HIV aid prior to PEPFAR, and others (all measured in 2004 baseline values; see Table 1).

Table 2 reports the descriptive statistics for all PEPFAR countries, the subset that are COP countries, and the comparison group countries. There are 90 PEPFAR countries in the database, of which 31 are COP countries. PEPFAR distributed aid to nearly half of the countries in the world over the period, comprising three-quarters of the global population. The average population size of PEPFAR countries is 62 million compared to 12.8 million for the comparison group countries.

## GDP per capita growth rate

Fig 1 shows the 1990–2018 trends in the GDP per capita growth rate for all PEPFAR countries, COP countries, and the comparison group countries. In general, the comparison group countries exhibit significantly greater variability than the PEPFAR countries over the entire period. From 2000 to 2004, the trends in GDP per capita are similar and vary within a narrow band—

**Table 2. Descriptive statistics–All PEPFAR, COP, and comparison group countries.**

| Variable | All PEPFAR Funded LMICs | COPs | Comparison Group (non-PEPFAR LMICs) |
|---|---|---|---|
| Number of countries | 90 | 31 | 67 |
| Total population 2018 | 5,609,546,475 | 2,680,309,948 | 860,246,053 |
| Cumulative PEPFAR disbursements 2004–2018 | $40,920,244,737 | $39,783,701,262 | $8,025,017 |
| Cumulative PEPFAR disbursements per capita, 2004–2018 | $3,094.20 | $2,974.10 | $0.40 |
| Cumulative other donor health aid per capita (non-PEPFAR donor + non-HIV donor+US), 2004–2018 | $9,428.40 | $3,805.50 | $4,525.90 |
| Cumulative health spending per capita (domestic) [2000–2016] | $365,066 | $85,261 | $588,474 |
| BL GDP/capita | $1,761.90 | $1,092.60 | $4,654.60 |
| BL HIV prevalence rate | 3 | 7 | 0.2 |
| BL life expectancy at birth | 61.1 | 55.1 | 71.3 |
| BL population urban | 41.70% | 33.80% | 58.00% |
| BL % adult. pop > primary education—secondary | 55.80% | 43.40% | 82.00% |
| No. countries receiving U.S. HIV aid before 2004 | 54 | 25 | 2 |
| BL fertility | 4 | 4.4 | 2.6 |

**Source:** Authors' tabulations of panel dataset.

Notes: BL = baseline; COP = Country Operational Plans; GDP = gross domestic product; HIV = human immunodeficiency virus; LMICs = low- and middle-income countries; PEPFAR = US President's Emergency Plan for AIDS Relief.

although not strictly parallel as required by the DID methodology. For all countries, growth rates peak in 2004 and then decline over most of 2004–2018, with a slight increase in growth in the PEPFAR countries at the end of the period. Although all countries continue to experience

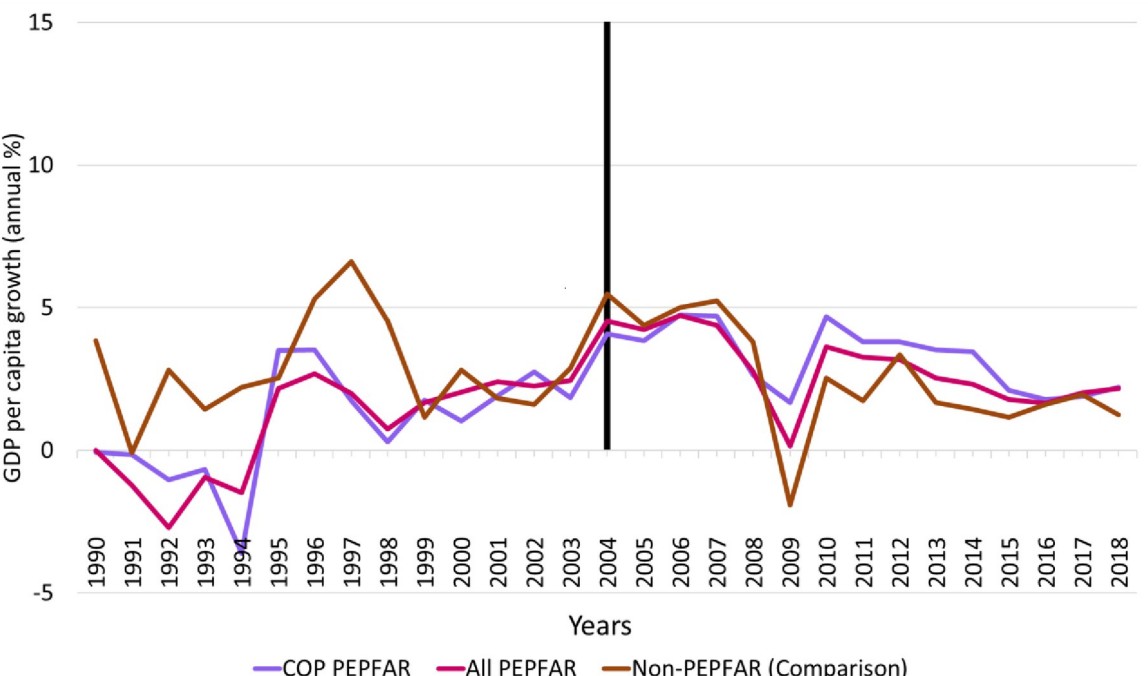

**Fig 1. Trends in GDP per capita growth rate, 1990–2018 for PEPFAR, COP and Non-PEPFAR countries. Source:** Authors' tabulations. Notes: Vertical line indicates the formal year of initiation of the PEPFAR program; COP = Country Operational Plans; GDP = gross domestic product; PEPFAR = US President's Emergency Plan for AIDS Relief.

GDP growth per capita, the rate of growth slows more in the comparison group countries than either the total PEPFAR group of countries or the COP countries. At around the time of the 2009 global recession, GDP per capita growth rates for all PEPFAR countries and COP countries began to exceed those of comparison group countries and remained higher throughout the remainder of the follow-up period. Although the PEPFAR program was formally introduced in most countries in 2004, it should be noted that efforts to address the HIV pandemic were underway in many countries prior to 2004. There is a substantial increase in GDP per capita in PEPFAR countries and comparison group countries over 2003–2004 but this is unlikely to be a result of HIV programmatic spending as growth rates were highest in the comparison group countries.

Table 3 reports the key DID model results for GDP growth per capita growth rate for all PEPFAR countries and COP countries relative to comparison group countries. Two sets of model results are reported: DID models that do not include any baseline control variables and those that do. In general, the unadjusted models closely mirror the descriptive trends in GDP among the different comparison groups. The PEPFAR ALL and PEPFAR COP parameter estimates measure the baseline differences in GDP per capita growth rate for these countries relative to the comparison group at baseline. The unadjusted model results indicate that GDP per capita growth rates were roughly 2 percentage points lower in both groups of PEPFAR countries than comparison group countries at baseline and these differences were highly significant statistically. The sign and significance of baseline differences were similar in the adjusted models. The parameter estimates for the TIME variable measure the trend in the comparison group relative to the baseline. Although the plot of GDP per capita growth rate shows fluctuation in the comparison group over time, there is no discernible trend, and the variable is statistically insignificant in both the adjusted and unadjusted models. Finally, the INTERACTION variable measures the impact of PEPFAR on GDP per capita growth rate. For the PEPFAR

**Table 3. DID models of GDP per capita growth rate.**

| GDP Growth | Unadjusted Model Total PEPFAR | Adjusted Model Total PEPFAR | Unadjusted Model COPs | Adjusted Model COPs |
|---|---|---|---|---|
| | Coefficient (t-statistic) | Coefficient (t-statistic) | Coefficient (t-statistic) | Coefficient (t-statistic) |
| Constant | 2.875*** | 7.691** | 2.875*** | 13.50*** |
| | (0.247) | (2.515) | -0.268 | -4.097 |
| Time | -0.287 | -0.112 | -0.287 | -0.125 |
| | (0.333) | (0.343) | (0.360) | (0.373) |
| PEPFAR | -1.977*** | -1.754*** | n/a | n/a |
| | (0.317) | (0.389) | n/a | n/a |
| PEPFAR COPs | n/a | n/a | -1.950*** | -2.100** |
| | n/a | n/a | (0.444) | (0.796) |
| Interaction—PEPFAR | 2.276*** | 2.072*** | n/a | n/a |
| | (0.429) | (0.434) | n/a | n/a |
| Interaction–COPs | n/a | n/a | 2.623*** | 2.504*** |
| | n/a | n/a | (0.610) | (0.615) |
| Adj. R-squared | 0.015 | 0.035 | 0.01 | 0.025 |
| N | 4,192 | 3,865 | 2,558 | 2,283 |

**Source:** Authors' analyses.

Notes

***p < 0.001

**p < 0.01. Adj = adjusted; COP = Country Operational Plans; DID = difference-in-difference; n/a = not applicable; PEPFAR = US President's Emergency Plan for AIDS Relief.

ALL group, the PEPFAR effect is positive and statistically significant. As expected, the magnitude of the PEPFAR impacts was highest for the PEPFAR COP countries (2.50 versus 2.07 for the broader group of PEPFAR countries in the adjusted models). It should be noted that the adjusted R-squares for all models are very low, indicating that PEPFAR explains a small amount of the variability in GDP per capita growth rate.

### Female primary school disengagement

Fig 2 reports the trends in female primary school disengagement for all PEPFAR countries, COP countries, and comparison group countries from 1990–2018. Baseline levels of disengagement are substantially higher in the PEPFAR countries relative to comparison group countries. There is a gradual improvement in female primary school disengagement rates in the comparison group countries over 1990–2004 but this appears to flatten out after 2004. Over the period 1997/98 to 2003, there is a steep improvement in female primary school disengagement rates in PEPFAR countries. Following the introduction of PEPFAR in 2004, rates of female disengagement for PEPFAR appear to converge toward the comparison group countries—particularly for PEPFAR countries that prepare COPs.

Table 4 reports the key unadjusted and adjusted DID model results for female primary school disengagement for all PEPFAR countries and COP countries relative to comparison group countries. The definitions of all key variables are the same as previously described for the GDP per capita growth models. The PEPFAR ALL and PEPFAR COP coefficients indicate very large differences in baseline levels of primary school disengagement for females relative to comparison group countries. The unadjusted model results indicate that rates of female primary school disengagement were more than 19 percent higher in all PEPFAR countries than in comparison group countries at baseline; the baseline differences were about 18 percent higher in COP countries. However, after controlling for baseline characteristics, these baseline differences were no longer statistically significant for the COP countries. The parameter

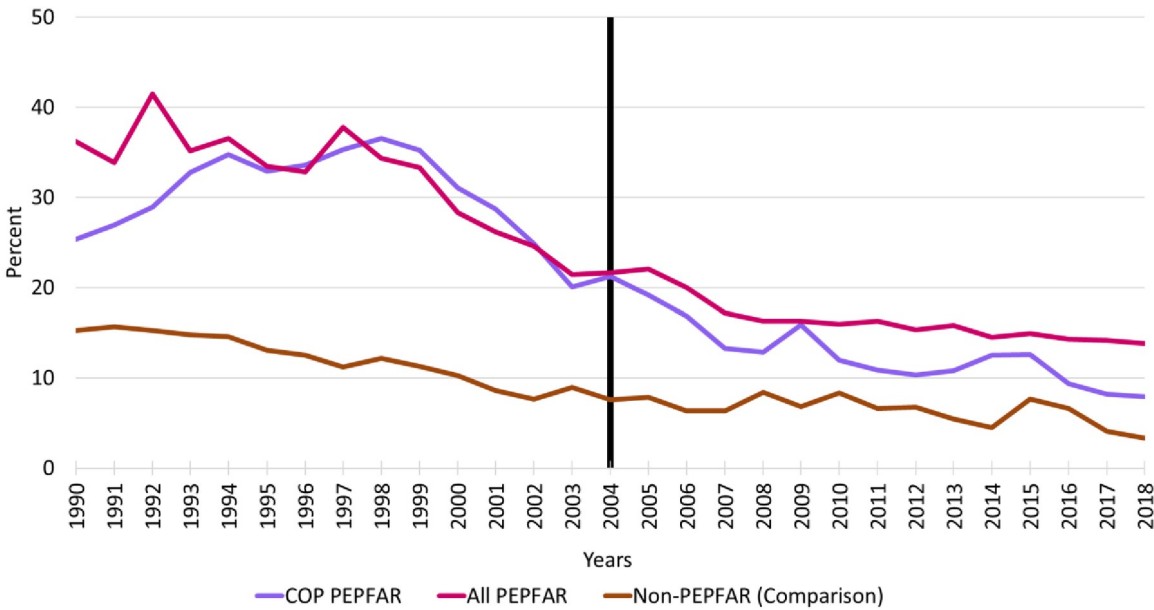

**Fig 2. Female primary school disengagement, 1990–2018. Source:** Authors' tabulations. Notes: Vertical line indicates the formal year of initiation of the PEPFAR program; COP = Country Operational Plans; PEPFAR = US President's Emergency Plan for AIDS Relief.

**Table 4. DID models of female primary school disengagement.**

| Female Education | Unadjusted model PEPFAR | Adjusted model PEPFAR | Unadjusted model COPs | Adjusted model COPs |
|---|---|---|---|---|
| | Coefficient (t-statistic) | Coefficient (t-statistic) | Coefficient (t-statistic) | Coefficient (t-statistic) |
| **Constant** | 11.54*** | 68.15*** | 11.54*** | 67.94*** |
| | (1.053) | (6.647) | -0.785 | -9.077 |
| Time | -4.915*** | -4.551*** | -4.915*** | -4.113*** |
| | (1.365) | (0.928) | (1.019) | (0.835) |
| **PEPFAR** | 19.37*** | 5.895*** | n/a | n/a |
| | (1.320) | (1.049) | n/a | n/a |
| **PEPFAR COPs** | n/a | n/a | 18.31*** | 1.598 |
| | n/a | n/a | (1.250) | (1.735) |
| **Interaction—PEPFAR** | -9.271*** | -9.185*** | n/a | n/a |
| | (1.713) | (1.143) | n/a | n/a |
| **Interaction–COPs** | n/a | n/a | -11.60*** | -12.58*** |
| | n/a | n/a | (1.658) | (1.304) |
| **Adj. R-squared** | 0.220 | 0.669 | 0.291 | 0.609 |
| **N** | 1,669 | 1,577 | 969 | 901 |

**Source:** Authors' Analyses.

Notes

***$p < 0.001$

**$p < 0.01$; Adj = adjusted; COP = Country Operational Plans; DID = difference-in-difference; n/a = not applicable; PEPFAR = US President's Emergency Plan for AIDS Relief.

estimates for the TIME variable show that, after controlling for other baseline variables, levels of disengagement in the comparison group trended upward by approximately 4–5 percentage points across the unadjusted and adjusted PEPFAR ALL and COP models from 2004–2018. Controlling for the baseline differences and trends in the comparison group countries, the INTERACTION variable measuring the impact of PEPFAR on female primary school disengagement in the adjusted models is large and highly significant for both the PEPFAR ALL group (-9.18 percentage points) and the COP group (-12.58 percentage points). The magnitudes of these treatment effects were similar in the unadjusted models. Moreover, the adjusted R-squares for the models range from 0.67 to 0.61, indicating that these models explain a significant amount of the variation female primary school disengagement trends.

## Male primary school disengagement, 1990–2018

Fig 3 reports the trends in male primary school disengagement for all PEPFAR countries, COP countries, and comparison group countries from 1990–2018. As with females, baseline levels of male disengagement were substantially higher in the PEPFAR countries relative to comparison group countries and displayed similar trends in the baseline and follow-up periods to those of females. Following the introduction of PEPFAR, rates of male disengagement for PEPFAR appear to converge toward the comparison group countries—particularly for PEPFAR countries that prepared COPs.

Table 5 reports the key DID model results for male primary school disengagement for all PEPFAR countries and COP countries relative to the comparison group countries. As with the models for females, the PEPFAR ALL and PEPFAR COP variables indicate very large differences in baseline levels of primary school disengagement for males relative to comparison group countries. The results from the unadjusted models indicate that rates of male primary

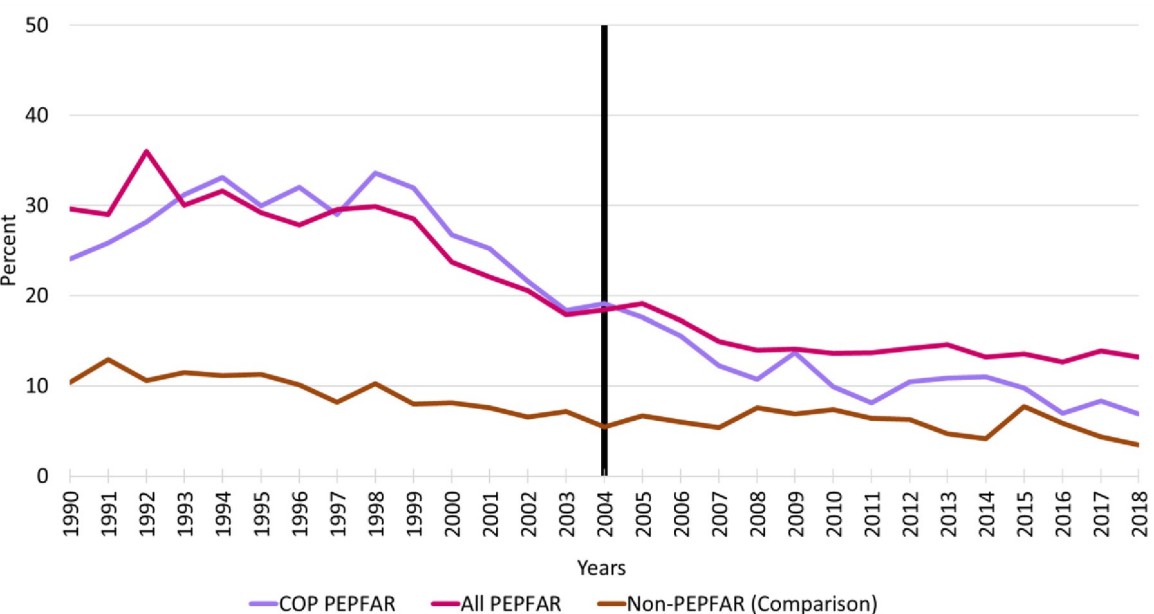

**Fig 3. Trends in male primary school disengagement.** Notes: Vertical line indicates the formal year of initiation of the PEPFAR program; COP = Country Operational Plans; PEPFAR = US President's Emergency Plan for AIDS Relief.

school disengagement were roughly 17 percentage points higher in all PEPFAR countries and 18 percentage points higher in COP countries at baseline. Not surprisingly, these differences were highly significant statistically. The parameter estimates for the TIME variable show that levels of disengagement in the comparison group decreased by roughly 3 percentage points from 2004–2018. Controlling for the trend in the comparison group countries, the INTERACTION variable measuring the impact of PEPFAR on male primary school disengagement was large and highly significant for both the PEPFAR ALL group (-7.96 percentage points) and the COP group (-12.51 percentage points). Moreover, the adjusted R-squares for the models range from 0.60 to 0.58, indicating that PEPFAR and the baseline control variables explain a significant amount of the variation in these trends.

## Trends in female employment rates

Fig 4 reports the trends in employment rates for females aged 15 and over. Although the baseline trends appear to be parallel, there is only a hint of an upward trend in the comparison group during the follow-up period while employment rates for women remained flat during this period. Despite no evidence of trends, it is apparent that employment rates for women are substantially higher in all PEPFAR countries and COP countries relative to comparison group countries.

The DID models for females are reported in Table 6. As anticipated based on the trends in Fig 4, the PEPFAR ALL and PEPFAR COP parameter estimates from the unadjusted models are 15.00 and 21.28, respectively, indicating large and statistically significant differences in female employment rates between the PEPFAR and comparison group countries over the baseline period. The TIME coefficient estimates from the unadjusted models indicate that comparison group employment increased by about 2.5 percentage points in the follow-up period relative to baseline. After controlling for baseline differences in female employment rates between the PEPFAR and comparison groups, the INTERACTION variable measuring

**Table 5. DID models of male primary school disengagement.**

| Male Education | Unadjusted model PEPFAR | Adjusted model PEPFAR | Unadjusted model COPs | Adjusted model COPs |
|---|---|---|---|---|
| | Coefficient (t-statistic) | Coefficient (t-statistic) | Coefficient (t-statistic) | Coefficient (t-statistic) |
| Constant | 9.054*** | 34.46*** | 9.054*** | 48.40*** |
| | (0.873) | (5.992) | -0.659 | -8.146 |
| Time | -3.010** | -2.863*** | -3.010*** | -2.577*** |
| | (1.132) | (0.837) | (0.855) | (0.749) |
| PEPFAR | 17.10*** | 7.023*** | n/a | n/a |
| | (1.094) | (0.945) | n/a | n/a |
| PEPFAR COPs | n/a | n/a | 18.12*** | 5.782*** |
| | n/a | n/a | (1.049) | (1.557) |
| Interaction—PEPFAR | -8.382*** | -7.962*** | n/a | n/a |
| | (1.420) | (1.031) | n/a | n/a |
| Interaction—COPs | n/a | n/a | -12.34*** | -12.51*** |
| | n/a | n/a | (1.392) | (1.171) |
| Adj. R-squared | 0.225 | 0.598 | 0.331 | 0.577 |
| N | 1,669 | 1,577 | 969 | 901 |

Source: Authors' analysis.

Notes

***$p < 0.001$

**$p < 0.01$ Adj = adjusted; COP = Country Operational Plans; DID = difference-in-difference; n/a = not applicable; PEPFAR = US President's Emergency Plan for AIDS Relief.

PEPFAR program impact is negative and statistically significant for both the PEPFAR ALL and COP groups. However, this does not seem to be due to a programmatic impact of PEPFAR but, rather, an upward trend in employment in the comparison group while employment rates remained unchanged in PEPFAR countries.

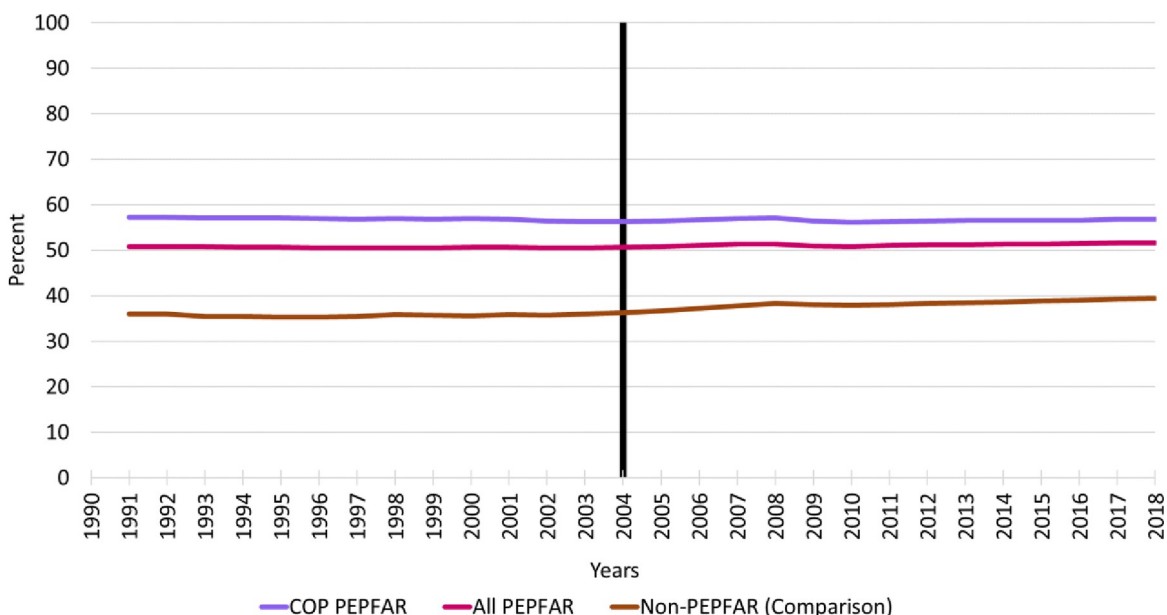

**Fig 4. Trends in female employment rates, 1990–2018.** Notes: Vertical line indicates the formal year of initiation of the PEPFAR program; COP = Country Operational Plans; PEPFAR = US President's Emergency Plan for AIDS Relief.

**Table 6. DID model of female employment rates.**

| Female Employment | Unadjusted model PEPFAR | Adjusted model PEPFAR | Unadjusted model COPs | Adjusted model COPs |
|---|---|---|---|---|
| | Coefficient (t-statistic) | Coefficient (t-statistic) | Coefficient (t-statistic) | Coefficient (t-statistic) |
| Constant | 35.64*** | 156.7*** | 35.64*** | 93.83*** |
| | (0.628) | (5.680) | -0.615 | -7.539 |
| Time | 2.484** | 2.869*** | 2.484** | 2.869*** |
| | (0.858) | (0.805) | (0.840) | (0.695) |
| PEPFAR | 15.00*** | 2.084* | n/a | n/a |
| | (0.790) | (0.885) | n/a | n/a |
| PEPFAR COPs | n/a | n/a | 21.28*** | 15.88*** |
| | n/a | n/a | (1.006) | (1.416) |
| Interaction—PEPFAR | -1.94 | -2.416* | n/a | n/a |
| | (1.080) | (0.991) | n/a | n/a |
| Interaction—COPs | n/a | n/a | -2.846* | -3.313** |
| | n/a | n/a | (1.374) | (1.102) |
| Adj. R-squared | 0.147 | 0.334 | 0.265 | 0.543 |
| N | 3,948 | 3,612 | 2,324 | 2,044 |

Source: Authors' analysis.

Notes

***$p < 0.001$

**$p < 0.01$

* $p < 0.05$. Adj = adjusted; COP = Country Operational Plans; DID = difference-in-difference; n/a = not applicable; PEPFAR = US President's Emergency Plan for AIDS Relief.

## Trends in male employment rates, 1990–2018

Fig 5 reports the trends in employment rates for males aged 15 and over. As with female employment the trends in the PEPFAR and comparison groups are basically horizontal lines over both the baseline and follow-up periods. In the follow-up period there is evidence of an a very modest trend in employment rates for males in the comparison group countries.

The DID model for male employment is shown in Table 7. The PEPFAR ALL and PEPFAR COP parameter estimates from the unadjusted models indicate that baseline employment levels were about 4.5 percentage points higher in PEPFAR ALL countries and 5.6 percentage points higher in COPs countries relative to the comparison group. There are no statistically significant employment trends for males in the comparison group countries. Finally, in the models that adjusted for baseline variables, the parameter estimates for the PEPFAR program INTERACTION term were statistically significant and negative for the ALL PEPFAR and COP countries. As with females, this appears to be due to a slight increase in employment in the comparison group while employment in PEPFAR countries remained flat over time.

## Sensitivity analyses

A large number of sensitivity analyses were conducted in addition to the main results reported here. Separate sets of models were estimated for the different PEPFAR country groups stratified by income classification (low and middle), as well as three five-year time periods. In general, the largest program impacts were observed for COP countries or countries where PEPFAR made the largest investments (which overlap significantly with COP countries). Results for these sensitivity analyses are reported in the Supporting Information materials (S1

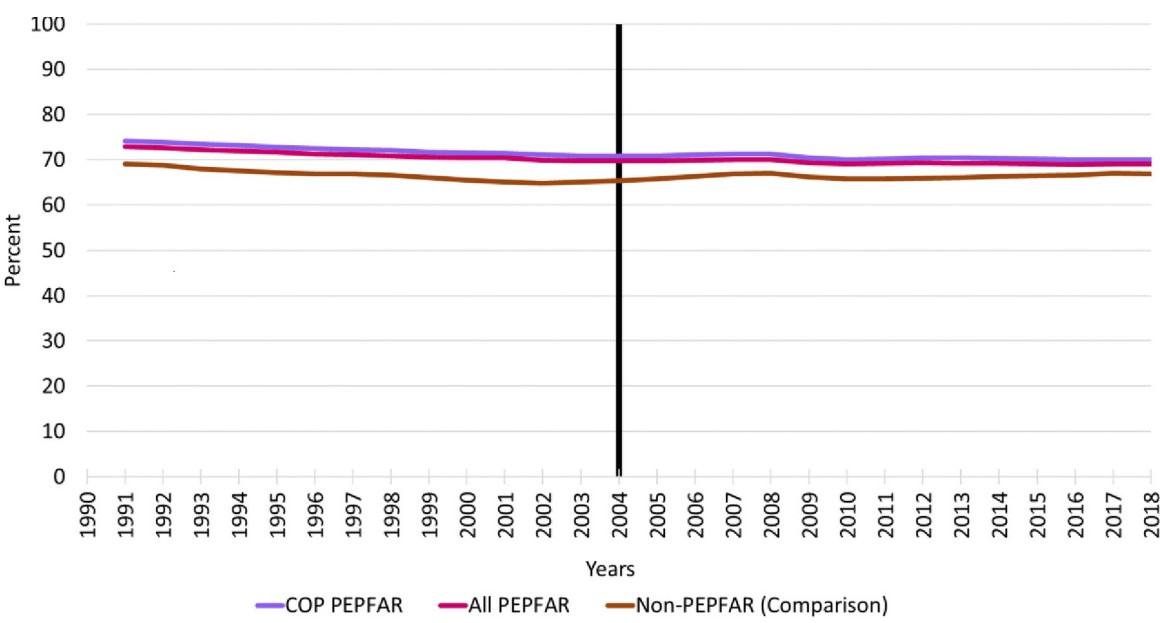

**Fig 5. Trends in male employment rates, 1990–2018.** Notes: Vertical line indicates the formal year of initiation of the PEPFAR program; COP = Country Operational Plans; PEPFAR = US President's Emergency Plan for AIDS Relief.

**Table 7. DID models of male employment rates.**

| Male Employment | Unadjusted model PEPFAR | Adjusted model PEPFAR | Unadjusted model COPs | Adjusted model COPs |
|---|---|---|---|---|
| | Coefficient (t-statistic) | Coefficient (t-statistic) | Coefficient (t-statistic) | Coefficient (t-statistic) |
| Constant | 66.72*** | 28.96*** | 66.72*** | -8.904* |
| | (0.413) | (3.578) | -0.383 | -4.405 |
| Time | -0.444 | -0.22 | -0.444 | -0.22 |
| | (0.564) | (0.507) | (0.523) | (0.406) |
| PEPFAR | 4.497*** | -0.557 | n/a | n/a |
| | (0.520) | (0.557) | n/a | n/a |
| PEPFAR COPs | n/a | n/a | 5.645*** | 5.338*** |
| | n/a | n/a | (0.627) | (0.827) |
| Interaction—PEPFAR | -1.389 | -1.657** | n/a | n/a |
| | (0.710) | (0.625) | n/a | n/a |
| Interaction—COPs | n/a | n/a | -1.451 | -1.650* |
| | n/a | n/a | (0.856) | (0.644) |
| Adj. R-squared | 0.031 | 0.314 | 0.055 | 0.460 |
| N | 3,948 | 3,612 | 2,324 | 2,044 |

Source: Authors' analysis.

Notes

***$p < 0.001$

**$p < 0.01$

* $p < 0.05$. Adj = adjusted; COP = Country Operational Plans; DID = difference-in-difference; n/a = not applicable; PEPFAR = US President's Emergency Plan for AIDS Relief.

Appendix). We also ran all models with and without China and India, the two most populous countries in the world, to assess whether they were influencing the results. In both cases, the results were similar. These results are also reported in the Supporting Information materials (S1 Appendix). Finally, we performed statistical tests for violations of the parallel trends assumption [28]. These tests indicated that the parallel trends assumption was not supported in some cases. Further research should be conducted to determine the sensitivity of the estimates in cases where the parallel trends assumption is violated.

## Discussion

This study confirms previous literature demonstrating that PEPFAR is associated with increases in economic growth [20,29], measured here by the GDP per capita growth rate. We show that these impacts are most pronounced in COP countries.

In addition, we demonstrate the impacts of PEPFAR on two measures not previously reported in the literature—girls' and boys' primary educational disengagement. PEPFAR was found to have large and statistically significant impacts on improving primary school engagement for both boys and girls. Again, the PEPFAR impacts were greatest in COP countries.

In contrast to a prior analysis [19] we do not find evidence for positive impacts of PEPFAR on rates of employment for females and males. Rates of employment for both females and males were essentially flat over the entire 1990–2018 period for the cohorts of all PEPFAR countries, COP countries, and comparison group countries. In the long run, it would be anticipated that reduced mortality as well as greater primary school educational engagement by both girls and boys should be reflected in higher rates of labor force participation and economic growth. However, such trends can take many years before they become evident. We consistently find that the positive macroeconomic externalities of PEPFAR on GDP growth and school engagement were the largest in COP countries, generally those that received the most money and engaged in intensive program planning.

## Limitations

Modeling the macroeconomic and educational spillover effects of PEPFAR is challenging due the complexity of the mechanisms through which substantial spending from a program like PEPFAR may work its way through a country's economy over time. For example, in addition to the potential impacts of health investments on mortality and morbidity, and subsequent impacts on labor supply and productivity, direct and indirect income effects of PEPFAR investments may contribute to aggregate demand. Investments in health care infrastructure generate income for health care workers which is then spent creating subsequent income for people working in other sectors. This income is then respent and, via Keynesian macroeconomic multipliers, generates potential benefits worth multiples of the original expenditure. Together, the demand and supply side effects generated by PEPFAR would be expected to have a positive impact on economic growth. In the long run, however, economic growth enhances the ability of a society to invest in further educational and health care infrastructure creating a positive feedback loop stemming from the original PEPFAR investment. Many of these issues are discussed by Piabuo and Tieguhong in their review of the literature on health expenditure and economic growth [4].

The differences-in-differences approach attempts to address these complexities by focusing attention on the program intervention itself. This requires making a strong assumption that the effects of factors not included in the model are fixed over time and are eliminated through the differencing procedure. Still, it is important to note that even with strong statistical

methods, estimation is challenging in the presence of feedback effects (e.g., better health results in greater economic growth and greater economic growth improves health).

## Supporting information

**S1 Appendix.** Exhibit A. Cohorts of PEPFAR countries created for analysis. Exhibit B. Country list by groups. Exhibit C. Difference-in-difference (DID) methodology. Exhibit D. Missingness. Exhibit E. Sensitivity tests for PEPFAR impact on GDP growth per capita. Exhibit F. Sensitivity test for PEPFAR impact on female primary school disengagement. Exhibit G. Sensitivity test for PEPFAR impact on male primary school disengagement. Exhibit H. Sensitivity test for PEPFAR impact on female employment rates. Exhibit I. Sensitivity test for PEPFAR impact on male employment rates.
(DOCX)

## Acknowledgments

The authors thank Adam Wexler and Stephanie Oum from KFF, Washington DC, for data retrieval and dataset preparation.

## Author Contributions

**Conceptualization:** William Crown, Jennifer Kates, Allyala Nandakumar.

**Data curation:** Dhwani Hariharan.

**Formal analysis:** William Crown, Dhwani Hariharan, Jennifer Kates, Gary Gaumer, Yiqun Luan, Allyala Nandakumar.

**Funding acquisition:** Allyala Nandakumar.

**Investigation:** William Crown, Jennifer Kates, Gary Gaumer, Allyala Nandakumar.

**Methodology:** William Crown, Yiqun Luan.

**Project administration:** William Crown, Monica Jordan.

**Software:** Yiqun Luan.

**Supervision:** William Crown, Allyala Nandakumar.

**Writing – original draft:** William Crown, Dhwani Hariharan, Jennifer Kates, Gary Gaumer, Yiqun Luan, Allyala Nandakumar.

**Writing – review & editing:** William Crown, Dhwani Hariharan, Jennifer Kates, Gary Gaumer, Clare Hurley, Yiqun Luan, Allyala Nandakumar.

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
