## [Decision Letter · Decision Letter 0]

2 Oct 2023

PONE-D-23-22009Analysis of Economic and Educational Spillover Effects in PEPFAR CountriesPLOS ONE

Dear Dr. Crown,

Thank you for submitting your manuscript to PLOS ONE. After careful consideration, we feel that it has merit but does not fully meet PLOS ONE’s publication criteria as it currently stands. Therefore, we invite you to submit a revised version of the manuscript that addresses the points raised during the review process.

Kind regards,

Ibrahim Jahun, MD, MSC, PhD

Academic Editor

PLOS ONE

Journal Requirements:

**Additional Editor Comments:**

The manuscript is well written and clearly illustrated the impact of PEPFAR beyond the primary objectives of the initiative over the years.  Please review references, tables and figures to ensure that they are all aligned with PLOS ONE’s publication criteria.

Reviewers' comments:

**Comments to the Author**

1. Is the manuscript technically sound, and do the data support the conclusions?

Reviewer #1: Yes

2. Has the statistical analysis been performed appropriately and rigorously? 

Reviewer #1: I Don't Know

3. Have the authors made all data underlying the findings in their manuscript fully available?

Reviewer #1: Yes

4. Is the manuscript presented in an intelligible fashion and written in standard English?

Reviewer #1: Yes

5. Review Comments to the Author

Reviewer #1: The manuscript is technically sound and has provided in-depth analysis and impact of PEPFAR beyond health. The contents may serve as great advocacy tool for continuous funding. Another area that this paper should have added to this analysis is OVC (orphans and vulnerable children) program. PEPFAR has substantially invested in OVC programs beyond what have been invested in DREAMS. This paper discuss DREAMS but not OVC. I recommend exploring how billions of dollars invested in OVC programs have impacted on some of the analysis provided in this paper.

6. PLOS authors have the option to publish the peer review history of their article (what does this mean?). If published, this will include your full peer review and any attached files.

Reviewer #1: No

---

## [Author Response · Author response to Decision Letter 0]

1 Dec 2023

Additional Editor Comments:

The manuscript is well written and clearly illustrated the impact of PEPFAR beyond the primary objectives of the initiative over the years. Please review references, tables and figures to ensure that they are all aligned with PLOS ONE’s publication criteria.

Thank you. Completed.

Reviewer #1: The manuscript is technically sound and has provided in-depth analysis and impact of PEPFAR beyond health. The contents may serve as great advocacy tool for continuous funding. Another area that this paper should have added to this analysis is OVC (orphans and vulnerable children) program. PEPFAR has substantially invested in OVC programs beyond what have been invested in DREAMS. This paper discuss DREAMS but not OVC. I recommend exploring how billions of dollars invested in OVC programs have impacted on some of the analysis provided in this paper.

Thank you for this comment. We have added a discussion of OVC to that of DREAMS to provide context for the possible programs within PEPFAR that may have led to the observed educational impacts on girls and boys in secondary school.

---

## [Editor Report · Decision Letter 1]

14 Dec 2023

Analysis of Economic and Educational Spillover Effects in PEPFAR Countries

PONE-D-23-22009R1

Dear Dr. Crown,

We’re pleased to inform you that your manuscript has been judged scientifically suitable for publication and will be formally accepted for publication once it meets all outstanding technical requirements.

Kind regards,

Ibrahim Jahun, MD, MSC, PhD

Academic Editor

PLOS ONE
---

## [Editor Report · Acceptance letter]

19 Dec 2023

PONE-D-23-22009R1 

PLOS ONE

Dear Dr. Crown, 

I'm pleased to inform you that your manuscript has been deemed suitable for publication in PLOS ONE. Congratulations! Your manuscript is now being handed over to our production team.

Kind regards, 

on behalf of

Dr. Ibrahim Jahun 

Academic Editor

PLOS ONE